# Niche overlap across landscape variability in summer between two large herbivores using eDNA metabarcoding

Eduard Mas-Carrió [1]*, Marcin Churski[2], Dries Kuijper[2‡], Luca Fumagalli[1,3‡]

**1** Department of Ecology and Evolution, Laboratory for Conservation Biology, Biophore, University of Lausanne, Lausanne, Switzerland, **2** Mammal Research institute, Polish Academy of Sciences, Białowieża, Poland, **3** Swiss Human Institute of Forensic Taphonomy, University Centre of Legal Medicine Lausanne-Geneva, Lausanne University Hospital and University of Lausanne, Lausanne, Switzerland

‡ These authors contributed equally as senior scientists to this work
* eduard.mascarrio@unil.ch

**Data Availability Statement:** All data used for this study are available in the Dryad repository at the following link: https://doi.org/10.5061/dryad. k0p2ngfbn.

## Abstract

Understanding the relationship between a species feeding strategy and its environment (trophic ecology) is critical to assess environmental requirements and improve management policies. However, measuring trophic interactions remains challenging. Among the available methods, quantifying the plant composition of a species' diet indicates how species use their environment and their associated niche overlap. Nevertheless, most studies focusing on herbivore trophic ecology ignore the influence that landscape variability may have. Here, we explored how landscape variability influences diet composition through niche overlap. We used eDNA metabarcoding to quantify the diet composition of two large herbivores of the Bialowieza Forest, red deer (*Cervus elaphus*) and European bison (*Bison bonasus*) to investigate how increasing habitat quality (i.e. higher abundance of deciduous forage species) and predation risk (i.e. density of wolf in the area) influence their diet composition and niche partitioning. Our findings indicate diet composition is non-homogeneous across the landscape, both within and between species. Red deer showed greater diet variability and lower niche overlap within species compared to bison. We detected a reduction of niche overlap for red deer with increasing predation risk, leading to more dissimilar diets, suggesting their feeding behaviour is affected by wolf presence. This correlation was not found for bison, which are rarely predated by wolf. Higher habitat quality was associated with higher niche overlap only within bison, probably due to their suboptimal feeding strategy as browsers. These results show the importance of integrating environment-induced diet variation in studies aimed at determining the landscape usage or niche overlap of a species.

## Introduction

Mammalian herbivore species, and more precisely ungulates, are important regulators of the structure and functioning of forest ecosystems [1–4]. Environmental factors such as plant composition, herbivore and predator densities, human activity or protection regime influence each species differently. They create a multivariate landscape of environmental influences that

**Funding:** This project was supported by a Swiss National Science Foundation grant to LF (nr. 310030_192512). EM-C was supported by a fellowship in Life Sciences (Faculty of Biology and Medicine, University of Lausanne). The funders had no role in study design, data collection and analysis, decision to publish, or preparation of the manuscript.

**Competing interests:** The authors have declared that no competing interests exist.

affect the species overall body condition, feeding strategy, impact on the vegetation and modify their trophic ecology, i.e., feeding relationship between a species and its environment [5–7]. Thus, understanding how environmental factors influence the trophic ecology of herbivore species is key to correctly assess their ecological needs and adjust management policies accordingly [8]. These factors can be used as an indicator to assess the status of a particular species in their environment and their potential impact [9,10].

Quantifying diet composition provides a proxy for measuring species trophic ecology. It reveals the plant species consumed, and has been used to study diet selection, diet overlap and niche partitioning between species [9–11]. Traditional methods for diet quantification include micro-histology of scats [12] or rumen content analyses [13], but both methodologies are very labour intensive and are increasingly replaced by DNA metabarcoding [14,15]. DNA metabarcoding is the PCR amplification of short but informative barcodes with universal primers and next generation sequencing (NGS) of DNA mixtures [16] allows the simultaneous identification of different species within communities. This technique can be applied using the plant DNA present in the scats of herbivores, and provides a quantitative approach of the relative presence of each plant in the diet composition of each individual herbivore. Compared to traditional diet quantification methods, DNA metabarcoding can handle many samples at once and provides detailed information on diet composition, unlocking a new path for community ecology to study ecosystem functioning. These dietary studies have the potential to unfold the relation between herbivore diet selection and the environmental variables of interest, providing a non-invasive tool for wildlife management to assess the habitat use of a species in their ecosystem and the role each environmental variable has.

However, very little is known on the linkage between environmental factors, species interactions and diet composition among herbivore species [17–20]. The difficulties arise mainly from the many environmental variables acting simultaneously at landscape scale on each individual and how they influence diet composition. To tackle this problem, a new concept has been recently proposed to differentiate landscape-scale herbivory regimes, i.e., integrate multiple environmental variables and how they are perceived by herbivores, the so-called "herbiscapes" [21]. In brief, each herbiscape combines all the environmental variables sampled throughout the forest, divided in landscape grid cells. Using hierarchical clustering, areas can then be categorized to represent ecologically distinct landscape scale herbivory regimes, i.e., herbiscapes. In the Bialowieza forest, these functional areas have been described and we aim to use the described spatial variability within environmental factors to study how they affect diet composition within and between herbivore species. Overall, this creates an interesting experimental setup, where two sympatric species have to deal with distinct combinations of environmental factors within the whole ecosystem. The herbiscape approach could be advantageous for applied community ecology studies because they condense complex multivariate spaces into distinct functional areas.

We used scat DNA metabarcoding to characterise in the Bialowieza Primaeval Forest the diet composition of two common ungulate species, red deer and bison. We tested the role of predation risk (imposed by wolf, *Canis lupus*) and habitat quality (i.e. proportion of deciduous forest and mean altitude) as environmental factors on the diet overlap for each species and highlighted the key discriminant plant species. We further assessed the role of the environmental variables on the niche overlap within and between the two species. We hypothesised: (i) Low predation risk and high habitat quality are associated with higher interspecies niche overlap, as both species will select high nutrient plant species in low risk areas (for example, [22]); (ii) Within species niche overlap is more sensitive to the environmental factors, but overall higher in bison due to their suboptimal feeding as browsers [23]; (iii) red deer behaviour

changes due to wolf predation and its niche overlap is reduced with increasing predation risk compared to bison, which remains unaltered as the species is rarely predated [21].

## Materials and methods

### Study area

Scat sampling was conducted in the Polish part of the Bialowieza Primaeval Forest (Fig 1A and 1B), in the north-western part of Poland, on the border with Belarus. The area has a total extension of 600 km$^2$ and covers a well-preserved fragment of the primaeval forest (Fig 1B), unique in Europe for its exclusion of forestry and ungulate management since 1921. The Polish part of the forest has two distinct management zones: the Bialowieza National Park (BNP, 105 km$^2$), without forestry nor ungulate management, and the state forest, with forestry and ungulate management. The Bialowieza forest is composed of different ecosystems, such as open grasslands, wet marshes and deadwood forests, but is mainly dominated by the primaeval forest. Deciduous forests are the dominant type and include hornbeam (*Carpinus betulus*), oak (*Quercus robur*) and lime (*Tilia cordata*). Tree stands in the state forest are dominated by spruce (*Picea sp.*) [24]. Herbivore species present include moose (*Alces alces*), bison (*bison bonasus*), roe deer (*Capreolus capreolus*), wild boar (*Sus scrofa*) and red deer (*Cervus elaphus*). Carnivore species present are wolves (*Canis lupus*) and lynx (*Lynx lynx*). The area is mainly known for hosting a free-ranging population of approximately 500 bison. In terms of wildlife management, the forestry department applies supplementary feeding in winter and culling of bison outside the park.

### Herbiscapes

Five distinct areas in terms of herbivory regime, i.e. herbiscapes, were defined for the Polish part of the Bialowieza forest (Fig 1C). Herbiscapes were calculated using camera traps, remote-sensing technologies, vegetation surveys, animal density inference, human activity patterns and other on-the-ground surveys (full list of variables listed in [21]). The forest was divided in 500 x 500m grid cells and all the sampled variables were then used for hierarchical clustering. The clustering assigned each of the forest grid cell based on their particular combination of environmental variables to one of the five distinct herbiscapes, i.e. the functional areas within the Bialowieza forest in terms of herbivory regime. Within each herbiscape, the specific areas for scat sampling were selected because they vary in terms of predation risk and habitat quality. Predation risk, one of the variables used to determine the distinct herbiscapes, was inferred based on the landscape use of wolf and lynx (using camera traps), driving the predator encounter rate by ungulates. In other words, the relative abundance of predators inferred from the camera trap survey was transformed into low, medium or high predation risk categories for each herbiscape in order to be comparable across areas. Habitat quality was defined as high or low based on the key biophysical properties defining forest composition, i.e. the percentage of landscape openness, canopy height and the percentage of coniferous species. For herbivores, these variables determine the resource availability and palatability of each forest grid cell. Moreover, the selected areas include high density of red deer and bison in each of them and in order to include also two areas inside the National park and three in the managed part. For both species, the GPS locations of the scat samples were considered to associate the predation risk and habitat quality categories to each sampled area (Table 1; for a full description, see [21]).

### Sample collection

Sampling of red deer and bison scats was done in June 2019. 10 fresh scat samples were collected per area and species, adding up to a total of 100 samples, 50 for each species. Samples

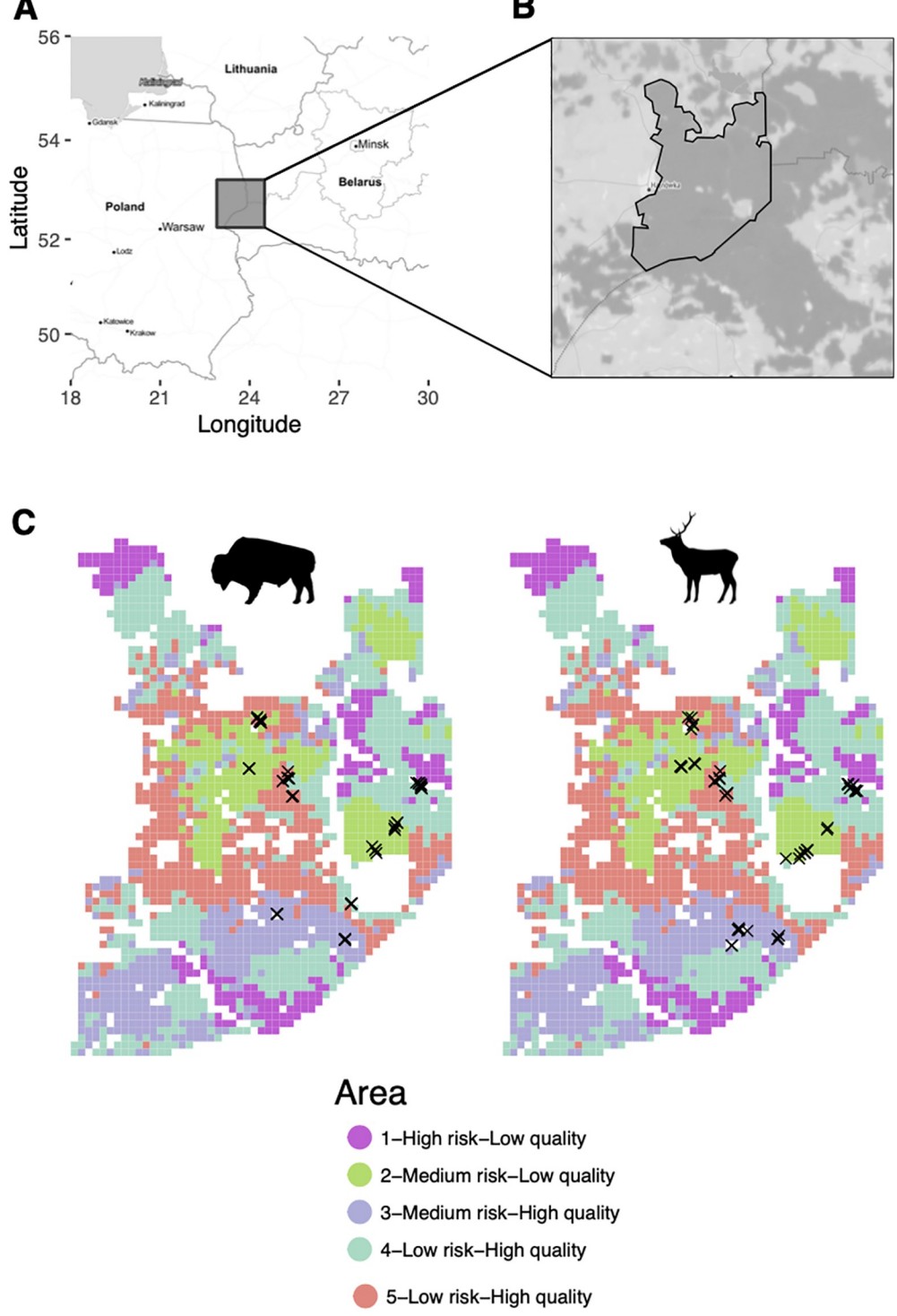

**Fig 1. Maps of the study area.** A) Large scale map of the area. Grey square marks the area where the Bialowieza forest is located. B) Dark grey areas indicate forested areas. Full black lines mark the area of study within the Bialowieza forest. Dashed black line indicates the limits of the Bialowieza National Park. Notice the eastern edge of the study area is limited by the border with Belarus. C) Area of study divided by the different herbiscape described in Bubnicki et al. 2019. We used the same herbiscape numbering. Each herbiscape has a unique and arbitrary color (herbiscape numbering is maintained as in [21]). Crosses indicate the location where each scat sample was collected for each species. Map tiles used in A) and B) were extracted from Stamen Design, under CC BY 4.0. Data by OpenStreetMap, under ODbL.

**Table 1. Herbiscape data used for the statistical analyses.**

| Herbiscape | Elevation | Forest type | Reserve status | Predation risk | Habitat quality |
|:---:|:---:|:---:|:---:|:---:|:---:|
| *1* | *Low* | *Mix (Coniferous)* | *National Park* | *High* | *Low* |
| *2* | *High* | *Coniferous* | *Managed forest* | *Medium* | *Low* |
| *3* | *Moderate* | *Deciduous* | *National park* | *Medium* | *High* |
| *4* | *High* | *Mix (Deciduous)* | *Managed forest* | *Low* | *High* |
| *5* | *Moderate* | *Deciduous* | *Managed forest* | *Low* | *High* |

were taken distant from each other and the GPS coordinates were recorded. The scat samples were collected fresh and stored in silica beads, in order to dry and preserve them without freezing until DNA extraction.

## DNA extraction

We used between 0.5 and 1 g of dry scat material as the starting point for the extraction. Extractions were performed using the NucleoSpin Soil Kit (Macherey-Nagel, Düren, Germany) following the manufacturer protocol. A subset of the extractions was tested for inhibitors with quantitative real-time PCR (qPCR) applying different dilutions (2x, 10x and 50x) in triplicates. Following these analyses, all samples were diluted 5-fold before PCR amplification. All extractions were performed in a laboratory restricted to low DNA-content analyses.

## DNA metabarcoding

DNA extracts were amplified using a generalist plant primer pair (Sper01, [16]), targeting all vascular plant species (see Supplementary Material for detailed laboratory procedure). Sper01 targets the P6 loop of the *trn*L intron (UAA) of chloroplast DNA (10–220 bp). To reduce tag jumps [25], we followed the library preparation as in [26]. Final libraries were quantified, normalised and pooled before 150 paired-end sequencing on an Illumina MiniSeq sequencing system with a Mid Output Kit (Illumina, San Diego, CA, USA).

## Bioinformatic data analyses

The bioinformatic processing of the raw sequence output and first filtering was done using the *OBITools* package [27] and is detailed in the Supplementary material. Remaining sequences were taxonomically assigned to taxa with a database for Sper01 (Supplementary material) generated using the EMBL database (European Molecular Biology Laboratory). Further data cleaning and filtering was done in R (version 4.0.2) using the *metabaR* package [28]. Remaining PCR replicates for each individual scat were then merged. The reads within each PCR replicate were pooled to calculate the relative abundance of each OTU (i.e. mean relative read abundance, RRA) and the proportion of PCR replicates of the same scat where each OTU was detected (i.e. PCR ratio, regardless of their relative abundance). We also grouped the OTUs into functional plant types (broadleaf, shrub, conifer, graminoid and forb).

## Statistics and modelling

All downstream analyses were carried out using R software (Version 4.0.2). Firstly, we calculated the dissimilarity matrix (Bray-Curtis distance) for each individual based on the final OTU table (transformed to relative read abundances, RRA) and visualised the dissimilarity between individuals using a non-metrical dissimilarity scaling (NMDS). Secondly, we quantified among-individual diet variation (V), i.e. the diet overlap between an individual and its

population within the *RInSp* package (V = 1 –*Psi*, [29]). It estimates the diet similarity in terms of plant OTUs between an individual and the average diet of its species. Values close to 0 indicate similar utilisation of resources whether values close to 1 indicate greater difference in diet composition.

Thirdly, we calculated Pianka's niche overlap index (Formula 1,[30]) using the *spaa* package to investigate niche overlap between individuals and across herbiscapes, which differ in terms of environmental variables, as follows:

$$Pianka_{jk} = \frac{\Sigma_i^n p_{ij} p_{ik}}{\sqrt{\Sigma_i^n p_{ij}^2 \Sigma_i^n p_{ik}^2}} \qquad \text{Formula 1}$$

where $p_{ij}$ and $p_{ik}$ are the proportion of plant OTU i by individual j and k, respectively, and n is the total number of plant OTU categories. Values close to 0 indicate no overlap, close to 1 indicates full overlap, i.e. same diets. Finally, we used generalised linear mixed models (GLMM), with the *glmmTMB* package, to investigate the significance of niche partitioning in terms of species interactions with herbiscapes and with the categorical variables associated with each herbiscape, i.e. predation risk and habitat quality. We first modelled the observed niche overlap against the three types of species interactions and the sampled areas (herbiscapes) using a beta distribution and with the *glmmTMB* R package, as follows:

$$\textit{Observed Niche overlap} \sim \textit{Species interaction} + \textit{Area} \qquad \text{Model 1}$$

We then tested niche overlap against predation risk and habitat quality. Both variables were significant in a linear mixed effect model, together with species interaction, which was included as an interaction variable. The models used was as follows (using a Beta family data distribution and with the *glmmTMB* R package) and held the best fit for the data with a meaningful combination of environmental variables:

$$\textit{Observed Niche overlap} \sim \textit{Species interaction} + \textit{Food quality} + \textit{Predation risk} + \textit{Species interaction} * \textit{Food quality} + \textit{Species interaction} * \textit{Predation risk} + (1|\textit{Area}) \qquad \text{Model 2}$$

We used the *performance* package [31] to assess which was the best distribution type for our data and compared multiple combinations of models. We retained the models with the highest marginal $R^2$.

## Results

### DNA metabarcoding

After all quality filtering steps, we retained 4,720,505 reads of 109 different OTUs (Operational Taxonomic Units) for the Sper01 assay that were assigned to 105 different taxa (1,373,406 for red deer and 3,347,099 for bison). Most relevant taxa were *Carpinus/Corylus sp.*, *Rubus idaeus* and *Quercus sp.*, which represent 80.8% of the sequences retained. OTUs grouped by plant type and herbivore species are shown in S1 Fig, where the sum of RRA of all individuals is shown by plant group.

### Red deer and bison diet composition

Bison clustered more than red deer for both axes of the NMDS, with both sharing a partial overlap in the multivariate space (Fig 2A). We visualised diet composition also by plant group for each herbiscape in order to compare them. Both species show a clear dominance of broadleaves in their diet for all the sampled areas, with bison having a greater broadleaf component

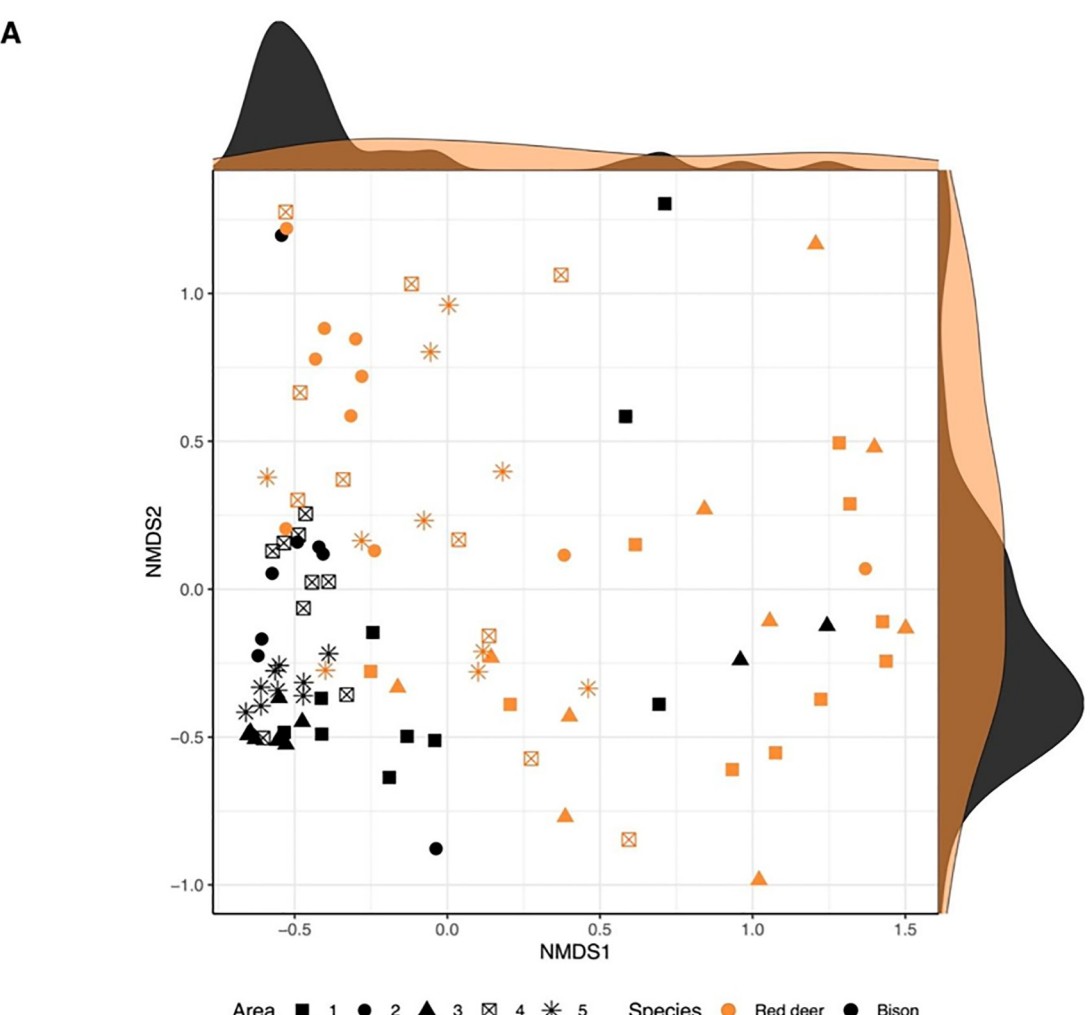

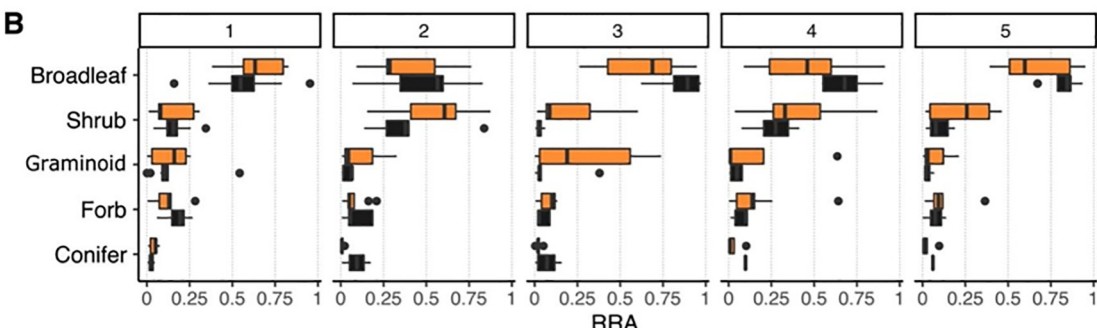

**Fig 2.** A) NMDS visualisation of the diet composition of all individuals. Density lines on the NMDS1 and NMDS2 axis indicate the density of points on each axis and each species. Dark density lines stand for bison and light grey ones for Red deer. B) Relative read abundance (RRA) for each plant type is shown separately for each species in each of the sampled areas.

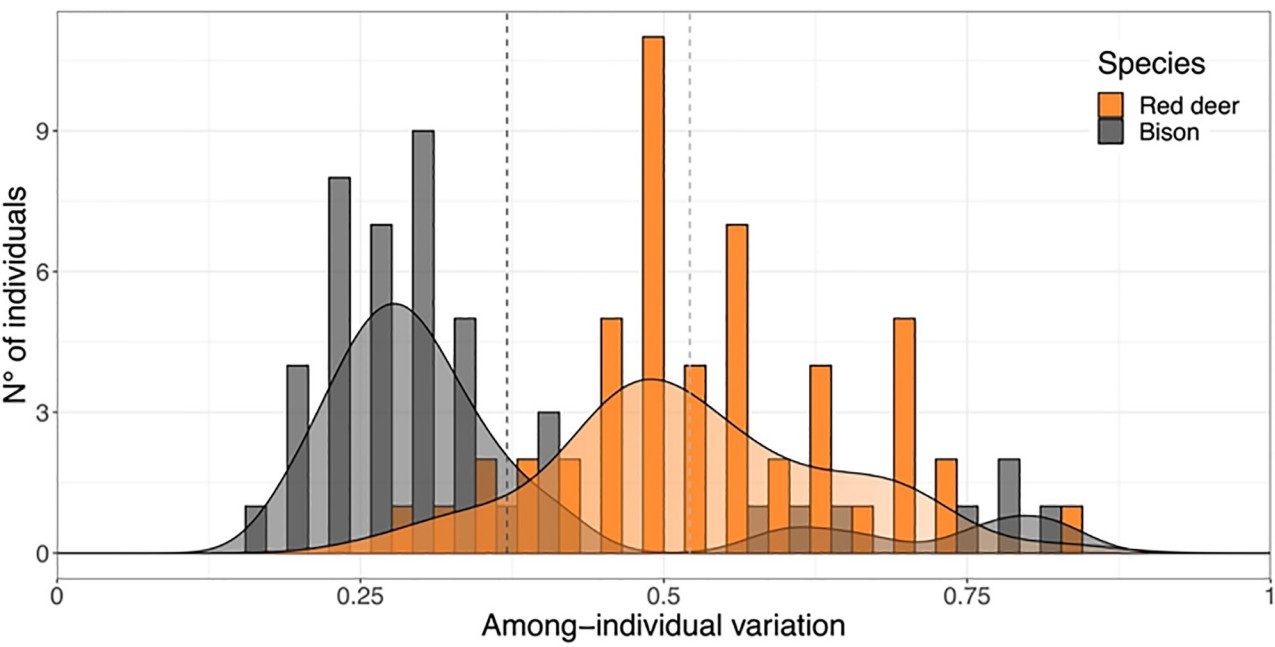

**Fig 3. Among-individual diet variation (V).** Dashed line indicates the mean value for each species.

in their diet compared to red deer (Fig 2B). Red deer register an overall higher contribution of the other plant groups compared to bison, except for conifers that was higher in bison.

## Among-individual diet variation

We found the two species to show a non-overlapping mean value of among-individual diet variation (V), with red deer having higher mean variation among individuals than bison (Fig 3). However, few bison individuals show higher among-individual diet variation, which indicates more diet variability compared to the majority of bison and will be investigated later on. In fact, diet clustering revealed that the majority of individuals grouped by species, except for 20 out of the total 100 individuals of both species, which clustered within the other species (S2 Fig, also for details).

## Niche overlap

We calculated the niche overlap (pianka's index, [30]) between each individual sample in order to test for the dietary niche overlap between and within species. In other words, we calculated bison intraspecific niche overlap, deer intraspecific niche overlap and deer-bison interspecific niche overlap. We kept only the comparisons within the same herbiscape, to exclude the potential differences in plant composition between areas. Model 1 highlighted differences in predicted niche overlap between species across the sampled areas, which reflects the variability within the forest ecosystem and suggests an interesting interplay of factors (Fig 4A). In line with Fig 3, niche overlap within bison was much higher than within red deer. We visualised both the estimates of each environmental variable on the observed niche overlap (Fig 4B and 4D) as well as the predicted niche overlap range calculated (Fig 4C and 4E) in Model 2. Habitat quality had the same effect within red deer and between red deer and bison, reducing niche overlap in high quality areas, contrary to within bison (Fig 4B and 4C). We detected an

increase in predicted niche overlap when predation risk is reduced, but this pattern is only clear within red deer (Fig 4D and 4E). Between species, high risk reduced niche overlap contrarily to low risk, but the overall niche overlap was similar across predation risk levels.

## Discussion

In this project, we studied the role of landscape variability on the diet of two large herbivores, in terms of plant species consumed and in terms of niche overlap between and within species. Areas were classified in five different categories or herbiscapes, which reflect different herbivory regimes across the landscape depending on the composition of the mammal community and other environmental variables. We used this framework to study the diet differences across the landscape between the two main ungulates of the forest, red deer and bison. We retrieved the plant composition of their diets and calculated their niche overlap. We tested these results against the different combinations of habitat quality and predation risk in order to investigate the relationship between herbivores' diet and the variability of the environment.

### Comparison of diet composition

The diet of both species was dominated by broadleaf plants (mainly *Corylus/Carpinus sp.*) during the sampled period (Fig 2B). These tree species, that abundantly occur in the area, produce new leaves during June and become the primary choice for both species, as previous studies have shown [23,32,33]. However, we found a higher proportion of all the other plant groups in red deer compared to bison (Fig 2B). In the NMDS visualization (Fig 2A), red deer individuals are distributed sparsely in the multivariate space compared to bison, which are more clustered (except few outlier individuals). This indicates deer samples have higher dietary differentiation compared to bison, but we cannot deduce from the NMDS visualization that red deer holds greater diet diversity of the overall population. In general, red deer are associated with eating a higher proportion of woody plant species (browsing), compared to bison preferring to forage on herbaceous plant species (grazing) [23,34]. According to the literature, bison prefers natural and human-made openings in the forest to profit from the grassy vegetation [35,36]. They have clear seasonal patterns with more woody vegetation in their diet during autumn-winter [23] and we should detect a greater proportion of graminoid species in their diet, which was not the case in our data (Fig 2B). This is an interesting finding as it provides a counterpoint to the extended idea that bison are not good browsers.

### Plant availability and animal movement

We were unable to study our dietary data in the light of the plant availability and their distribution within the forest. Having such data would have enabled to test the selectivity for each plant species and inform at a much finer level on the behavioural differences between the two studied herbivores. Certainly, the sampled scats reflected the whole feeding area and not just the immediate surroundings. Red deer and bison exploit differently their feeding grounds, and this have an impact on their feeding behaviour and dietary diversity, but we did not collect the data necessary for such approach, i.e. their movements record. We advocate future studies could perform fine scale dietary surveys accounting for the movement of individuals and local plant availability and distribution, but we also acknowledge the difficulty and volume of fieldwork required to retrieve the data.

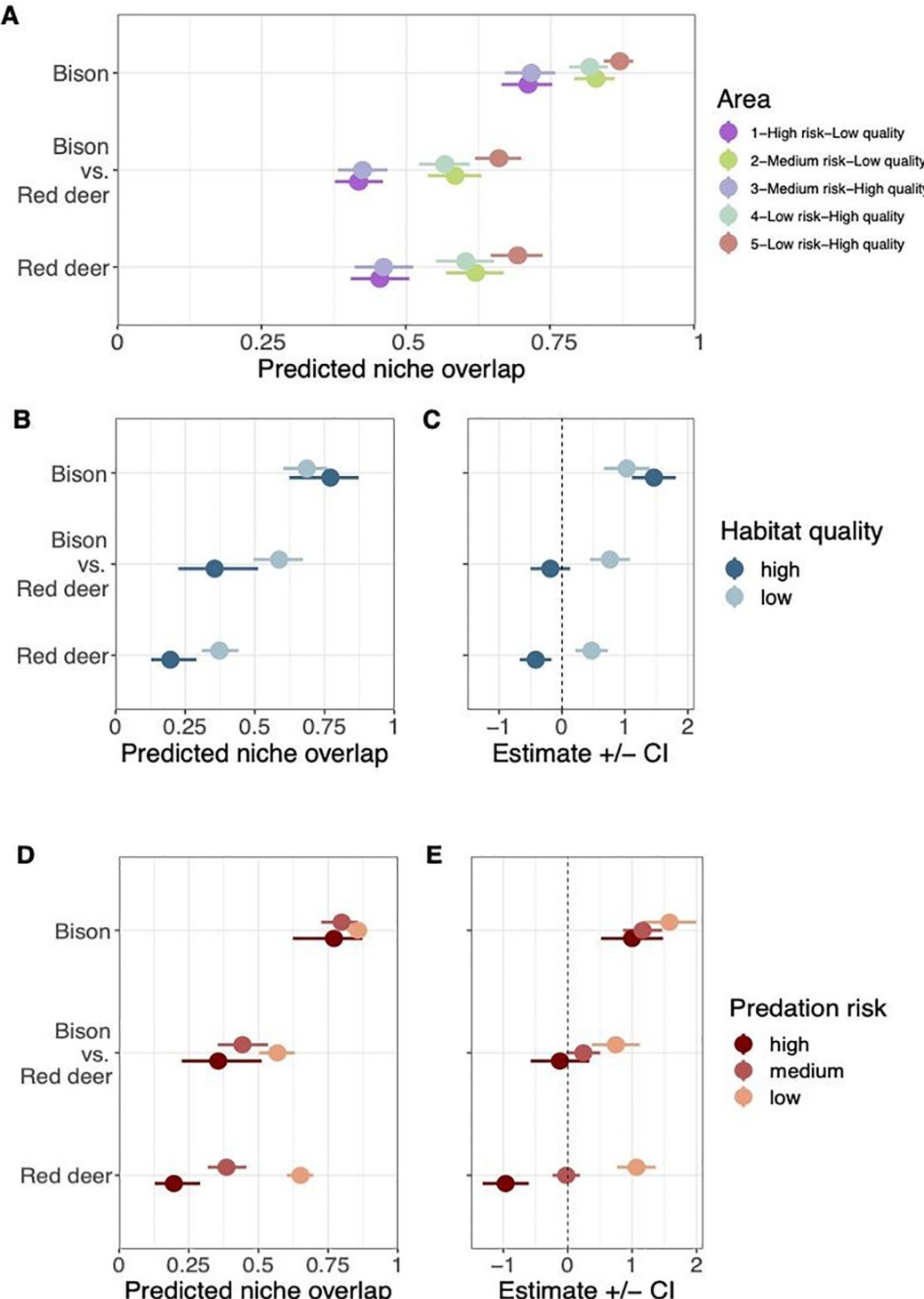

**Fig 4. Niche overlap modelling for each type of species interaction.** A) Predicted niche overlap by area (herbiscape). B) Predicted niche overlap for each habitat quality level. C) Estimates of the model for each species interaction and habitat quality. Dashed line indicates estimate value of 0. D) Predicted niche overlap for each predation risk level. E) Estimates of the model for each species interaction and predation risk.ashed line indicates estimate value of 0.

## Among-individual diet variation

Overlap in the diet composition between individuals and the average within species provides a straightforward method to assess the intraspecific variation in diet and thus can inform on the dietary plasticity of a species in their environment. Mean dietary dispersion was higher for red deer than for bison (Fig 3), i.e. red deer have higher diet variation within species than bison, in line with the NMDS (non-metrical dimensional scaling, Fig 2A), where greater dispersion of individuals means greater dissimilarity across their diets. In terms of their feeding strategy, we consider that the lower mean diet diversity of bison aligns to their grazer origin, making them less selective than typical browsers, i.e. red deer [37,38]. This could be explained because bison have wider muzzles, making them less selective when grazing or browsing. Alternatively, their larger digestive system could be homogenising the plants consumed and lead to more similarity between individual diets. However, we did not detect a clear signal of meadow plant species when comparing the two species (we did not observe any clear differences in graminoid proportion between the diet of both species, as seen in Fig 2B). As mentioned before, this result indicates that bison are not as dependent of grasses found in meadows for their summer diet. Red deer browse forest gaps with high regeneration growth [39,40], which are heterogeneously spread in an old forest like the Bialowieza forest. This implies covering more distance than grazers giving them access to a greater variety of resources [37]. Taking in account the differences between sampled areas, this result points to a greater pool of preferred plant species for red deer compared to bison, which could explain the mean diet diversity results.

## Niche overlap

**Intraspecific and interspecific interactions.** We observed bison to have the highest niche overlap within species, regardless of the sampled areas. Red deer niche overlap within species was clearly lower than bison and aligned to the interspecific niche overlap. The three areas outside the National park (2,4,5) show the highest niche overlap between species, i.e. most overlapping diet composition between bison and red deer. This could be related to behavioural similarities between the two herbivores. However, we cannot rule out differences in habitat composition outside the reserve, exploited for timber, which might offer a lower variety of plant species for the two species.

We associate the overall high niche overlaps within both species, particularly for bison, to the availability of food plants in the area. Despite we did not explicitly measure them, the role of resource abundance and palatability has to be considered when interpreting niche overlap. For instance, if there are few palatable plant species available, it is likely there will be higher overlap regardless of the total plant abundance. In our study, the abundance and diversity of resources available in spring probably exceeds the food requirements of the bison and red deer community in all the sampled areas and complicates the distinction between selectivity and competition. Hence, niche overlap is likely to align closer to competition during periods of limiting resources, i.e. winter. Thus, extending the experimental setup presented in this study to a year-round survey could reveal the seasonal dynamics between niche overlap and competition.

**Niche overlap and habitat quality.** In terms of habitat quality, high quality areas show greater niche overlap within bison but not within red deer and between species (Fig 4B). Both estimates of the model were positive, which suggests the differences in habitat quality does not affect the niche overlap within bison. This could be explained by the higher availability of their preferred plant species for bison in high quality areas, and contradicts our first hypothesis, which stated higher interspecies niche overlap would occur at low predation risk and high habitat quality, with both species selecting high nutrient plant species. Interestingly, red deer

estimates diverged compared to bison, as the model indicates low habitat quality increases the predicted niche overlap within species, contrary to high habitat quality (Fig 4C). The higher diversity in diet composition of red deer comes into play to explain this pattern (Fig 3). More likely, the browsing behaviour of red deer provides a fine-scale choice of plant species. Our data indicate red deer have more similar diet choices in lower quality areas. We detected a significant effect of low habitat quality to increase niche overlap, i.e., red deer has its niche overlap increased in low quality areas (Fig 4B and 4C). This could be due to red deer selecting for the same (and more palatable) plants, thus increasing their niche overlap. However, the plant composition of the environment or the competition between individuals could be also be playing a role. Between species, the model shows the same pattern as within red deer (Fig 4B and 4C). This suggests more diet convergence in low quality areas compared to high quality areas, where the greater plant diversity allows for both species to be more selective. We suggest that in high quality areas the greater diversity of plant species translates into lower predicted niche overlap between both species, as they feed on their preferred plant species. Moreover, the similarities between bison-red deer and red deer-red deer niche overlap confirms the facultative browsing nature of bison. Despite the many meadows surrounding the forest, our data suggest that bison spend most of their feeding time browsing within the forest, rather than grazing on the open meadows [23]. Grazing bison would reveal as a low niche overlap with red deer, which is not the case and aligns to our second hypothesis, i.e., bison have higher within species niche overlap due to their feeding behaviour. This could be confirmed by crossing our results with GPS data monitoring bison habitat use and test if niche overlap within bison correlates with the proportion of time spent on meadows. Our results show higher niche overlap variability within than between species, but they are not conclusive enough to answer if interspecific interaction can be used by management authorities as a fine-scale tool to measure the role of the environmental conditions.

**Niche overlap and predation risk.** We observed a complex relation with predation risk imposed by wolf and lynx on the two herbivores in terms of niche overlap, as firstly investigated in [34]. Our study covers more areas and more variables interplaying along the landscape, but measuring accurately these indirect effects using dietary data remains a challenge (see [8] for an overview on the advances in measuring predator-prey interactions). Between species, we detected a negative effect of predation risk on the diet overlap in high risk areas and positive in low risk areas (Fig 4E). This trend was more pronounced when comparing diet overlap between species than within species. However, the three types of species interaction converged in similar predicted niche overlap (Fig 4D), contrary to our second hypothesis, and illustrates the complex interplay of factors in the studied system. If both species were influenced by predation risk, we would have observed niche overlap increasing together with predation risk, which is not the case (Fig 4D). Such confounding results could be explained by the disparity in predation risk perception between red deer and bison, which modify the diet choice of red deer but does not affect the diet choice of bison, resulting in opposite estimates in the model but not in the predicted niche overlap.

Within bison, the higher niche overlap was kept across predation risk levels. The positive estimates detected for all predation risk levels (Fig 4E) is likely to be an artefact of the dietary choices, i.e. match between diet preference and plant availability, rather than a consequence of predation risk since bison is rarely predated. Their diet choice is only driven by selectivity and availability, and we can use them as a reference diet to assess the role of predation risk on red deer.

Red deer are highly predated by wolves and we hypothesized higher predation risk would result in lower within species niche overlap. In line with our third hypothesis, we found red deer had lower niche overlap within species with increasing predation risk, in contrast to

bison, which is rarely predated, and supports our model results. The negative estimate for high risk and positive for low risk (Fig 4E) indicates predation risk is an active factor driving red deer feeding strategy. We suggest these results respond to red deer reducing their selectivity and thus have a broader diet composition, leading to more dissimilar diets. The niche overlap in this species could be thus used as a tool to infer predation risk in the sampled area but also to estimate the abundance of predators, or at least the perception of predator abundance by red deer. This is an ambitious goal since other disturbances are likely to influence predation risk perception. We advocate this approach should be explored more in detail as we believe it has great potential to be used in wildlife management.

## Conclusion

This study shows eDNA dietary metabarcoding can be used to study herbivores' diet variation over the landscape and how diet is affected by environmental factors and interaction with other herbivore species. The integration of this technique in ecology studies will provide a new pathway to answer complex ecological questions. Moreover, this approach yields great potential to serve as a complementary tool for wildlife monitoring and species assessment in natural environments. Through diet composition, it can bring useful information for conservation purposes on herbivores' habitat use and feeding interactions. However, it requires an exhaustive characterization of the landscape so the diet composition of the study species can be used as a proxy to monitor the habitat use of species in their environment. We advocate future studies to explore this direction, but more comparative studies should be designed to assess the pros and cons of combining landscape ecology and dietary metabarcoding before drawing conservation and management policies using this methodology.

## Supporting information

**S1 File. Sdb.** Sper01 reference database.
(19)

**S2 File. Supplementary material.**
(DOCX)

**S1 Fig. Barplot by plant type.** Barplot by plant type and species, all individuals summed.
(PNG)

**S2 Fig. Cluster dendrogram.** Cluster dendrogram of diets (de = deer, bi = bison). Clustering analysis group species by diet composition similarity in a tree-like visualisation. If the first split of branches is grouped by species, all diets within species are more similar between each other than to any of the other species' individual diets.
(PNG)

## Acknowledgments

We thank J. Bubnicki for his help with the making and interpretation of herbiscapes raster map. We thank M. Baur and C. Stoffel for their help with the laboratory work and P. Becciu for the assistance with the modelling part. We also thank the Forest Research Institute in Białowieża for their support with the sampling process and the Białowieża National Park for granting the permits to access the reserve.

## Author Contributions

**Conceptualization:** Eduard Mas-Carrió, Marcin Churski, Dries Kuijper, Luca Fumagalli.

**Data curation:** Eduard Mas-Carrió.

**Formal analysis:** Eduard Mas-Carrió.

**Funding acquisition:** Luca Fumagalli.

**Investigation:** Eduard Mas-Carrió, Marcin Churski, Dries Kuijper.

**Methodology:** Eduard Mas-Carrió.

**Project administration:** Marcin Churski, Dries Kuijper, Luca Fumagalli.

**Resources:** Marcin Churski, Dries Kuijper, Luca Fumagalli.

**Software:** Eduard Mas-Carrió.

**Supervision:** Marcin Churski, Dries Kuijper, Luca Fumagalli.

**Validation:** Eduard Mas-Carrió, Dries Kuijper.

**Visualization:** Eduard Mas-Carrió.

**Writing – original draft:** Eduard Mas-Carrió.

**Writing – review & editing:** Eduard Mas-Carrió, Marcin Churski, Dries Kuijper, Luca Fumagalli.

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
