## [Decision Letter · Decision Letter 0]

15 May 2023

PONE-D-22-34041Niche overlap across landscape variability between two large herbivores using dietary eDNA metabarcoding in the Białowieża Primeval forest (Poland)PLOS ONE

Dear Dr. Mas-Carrio,

Thank you for submitting your manuscript to PLOS ONE. After careful consideration, we feel that it has merit but does not fully meet PLOS ONE’s publication criteria as it currently stands. Therefore, we invite you to submit a revised version of the manuscript that addresses the points raised during the review process. I believe the authors should address the limitations of their research and perhaps reshape a resubmission based on comments from reviewers, especially concerns by reviewer #1. I will detail a few suggestions below to assist in a possible revision:

1. Feces were only sampled in 1 month (June 2019) thus it seems the title and hypotheses could be edited to focus more on what you were actually able to test given the limitations of your sampling schedule and eDNA of samples collected.

Recommended title: Niche overlap across landscape variability during summer between two large herbivores using dietary eDNA metabarcoding

2. It is not clear why the models are presented in the Results instead of the Methods?

3. Please remove V and replace with "among-individual diet variation." It does not seem like it is necessary to reference this by V each time it is presented and I am also not sure why it is defined twice (Methods and Results).

4. As mentioned by both reviewers, predation risk needs more detail and justification. It appears that predation risk is based solely on the herbiscapes but the reader should not have to read that to determine predation risk. Instead of a half sentence (Line 147), more details on how "Predation risk was inferred based on landscape use..." is needed. Otherwise, the validity of your assessment needs to be determined based on reviewing Bubnicki et al. 2019.

We look forward to receiving your revised manuscript.

Kind regards,

W. David Walter, Ph.D.

Academic Editor

PLOS ONE

“This project was supported by a Swiss National Science Foundation grant to LF (nr. 310030_192512). EM-C was supported by a fellowship in Life Sciences (Faculty of Biology and Medicine, University of Lausanne).”

“This project was supported by a Swiss National Science Foundation grant to LF (nr. 310030_192512). EM-C was supported by a fellowship in Life Sciences (Faculty of Biology and Medicine, University of Lausanne). We thank M. Baur and C. Stoffel for their help with the laboratory work and P. Becciu for the assistance with the modelling part. We also thank the Forest Research Institute in Białowieża for their support with the sampling process and the Białowieża National Park for granting the permits to access the reserve.”

“This project was supported by a Swiss National Science Foundation grant to LF (nr. 310030_192512). EM-C was supported by a fellowship in Life Sciences (Faculty of Biology and Medicine, University of Lausanne).”

5. Please amend either the title on the online submission form (via Edit Submission) or the title in the manuscript so that they are identical.

6. We note that [Figure 1] in your submission contain [map/satellite] images which may be copyrighted. All PLOS content is published under the Creative Commons Attribution License (CC BY 4.0), which means that the manuscript, images, and Supporting Information files will be freely available online, and any third party is permitted to access, download, copy, distribute, and use these materials in any way, even commercially, with proper attribution. For these reasons, we cannot publish previously copyrighted maps or satellite images created using proprietary data, such as Google software (Google Maps, Street View, and Earth). For more information, see our copyright guidelines: http://journals.plos.org/plosone/s/licenses-and-copyright.

a. You may seek permission from the original copyright holder of Figure(s) [#] to publish the content specifically under the CC BY 4.0 license.   

Natural Earth (public domain): http://www.naturalearthdata.com/.

Reviewers' comments:

Reviewer's Responses to Questions

**Comments to the Author**

1. Is the manuscript technically sound, and do the data support the conclusions?

Reviewer #1: Yes

Reviewer #2: Partly

2. Has the statistical analysis been performed appropriately and rigorously? 

Reviewer #1: Yes

Reviewer #2: Yes

3. Have the authors made all data underlying the findings in their manuscript fully available?

Reviewer #1: Yes

Reviewer #2: Yes

4. Is the manuscript presented in an intelligible fashion and written in standard English?

Reviewer #1: Yes

Reviewer #2: Yes

5. Review Comments to the Author

Reviewer #1: The manuscript aims at assessing diet composition and niche overlap within and between red deer and European bison, and correlating such information with habitat quality and predator density, to support management actions. The approach is sound and mostly well explained, I just miss a comparison of the results with information on plant availability in the area and the inclusion of the species movement abilities while discussing the results.

General comments

• I honestly feel that using this approach to estimate predator abundance is an over ambitious goal, since other disturbances and the overall landscape of fear could influence this result. Talking of perception of predator density from the prey species is acceptable, but it should be discussed mentioning the possibility that other factors could also contribute to such perception.

• I missed a parallel with the observed diet composition and plants available in the areas considered. eDNA cannot be used to infer proportions in the diet, however, detecting abundant or scarce species in the diet would provide valuable information in this context.

• The species movement should be considered when discussing the results, as scats are not deposited where consumption occurred.

Specific comments

L36, L109, Fig.1: red deer instead of Red deer.

L65-66: this aspect could be better highlighted, since diet composition influences the overall body condition of the species, but also its impact on the vegetation.

L82: probably here the authors meant “study areas”.

L117: Probably better to replace “It” with “The area” or something alike.

L121: without forestry nor ungulate management.

L122: Considering the previous sentence mentions the whole Polish part of the forest, the BNP and the state forest, I’d suggest to replace “It” with a more explicit subject here.

L149: It seems like “both” refers to predation risk + habitat quality here, but I think the authors likely meant both species, please clarify.

L154-156: Where steel beads used in this process? Usually, nitrogen is added when working with plant material, so I was wondering if the authors used the steel beads as recommended instead.

L162-168: the lab procedure is sound and sufficiently explained, I would just suggest to add a reference to the supplementary information and specify the information on the Qubit Kit.

L171-172: Please specify the filtering criteria and add a reference to the supplementary information.

L174: the filtering in this step is clearly explained and sound.

L176-177: this sentence might be a bit challenging for readers with no previous experience with the method, I’d suggest making it a bit more beginner friendly.

L205: Please break down these values per species as well.

Fig.1: is not colour blind friendly. Additionally, please use a larger font for the x-axis.

L212-216: please briefly mention the differences among areas as well.

Supplementary Fig. 3: Please add test values.

L237-238: very good point!

L279: “in summer” is not really necessary here.

L283-284: This is consistent with NMDS results.

L285-288: as mentioned elsewhere, this statement is not fully supported by your results, so it should be better addressed here. Or you could move lines 301-304 here.

L304: could also be playing a role.

Reviewer #2: The authors conduct an interesting study of bison and red deer diets in the Bialowieza ecosystem. They find interesting associations between diet and ecosystem variables and they use appropriate metabarcoding methods. I believe the authors should consider their terminology in some places and consider the value of using both V and overlap. I also believe this is a good study and can be appropriate for publication in PLoS ONE after revisions.

Comments

Abstract: I do not believe that intraspecific dietary dissimilarity is an adequate indicator of predation risk; I think this section of the Abstract should be dramatically tempered. Without a comprehensive survey of possible contributing factors to red deer dietary dispersion (and wolf risk being the only strong predictor of it), it would be disingenuous to recommend that ecosystem managers solely need to measure red deer diets to quantify predation risk.

L73: there is a parenthesis missing

L89-98: please add a description of what the herbiscapes are in Bialowieza so the reader has an understanding of Bubnicki et al's results without needing to look up their paper; adding this will help center the reader on your study system and understand what factors go into herbiscapes. Please also add a citation to the Bialowieza-specific study

Throughout: I am also a bit uncomfortable with using niche overlap to describe intraspecific diet similarity. I realize that the author's use Pianka's overlap to quantify this so it technically is niche overlap, but I wonder if it might be easier to term the intraspecific values something like "dietary similarity" and reserve the term "niche" for the interspecific comparisons.

Methods: The section "Herbiscapes" needs to come before the section "Sample collection" because on L135 the authors state "per area per species" yet the reader has not been told that there are five areas yet

DNA metabarcoding methods: What was the told number of reads per library? It seems quite strict to remove sequences with <100 reads when many studies either remove only singletons or those with <10 reads

Methods: why use V and overlap for within species comparisons? Don't they show the same thing?

Fig.2: please add some color to the NMDS; it will make it much easier to distinguish the species

L227: "Food quality" should be "Habitat quality", right?

L227: The model structure is not completely clear to me; are the categorical predictors nested within species? Is this like having the categorical predictor as a fixed effect and per-species random intercepts?

L238-239: there is only two types of species interactions here (potential competition/facilitation between deer and bison; herbivory between the two species and their food plants). I can guess that the authors are referring to bison intraspecific overlap, deer intraspecific overlap, and deer-bison overlap. If I'm right, please rewrite this part to state that clearly, currently it is not clear.

Modelling methods: I'd like some more justification about the modelling approach for niche overlap. Why include "species interaction" as a predictor rather than just having three separate models (one for each species' intraspecific comparisons and one for interspecific overlap)? This will make it easier to interpret model output. Given that Area is just a proxy for predation risk and habitat quality, why not test those directly from the start (with Area as a random intercept)? It doesn't seem like there is much need for the Overlap ~ Area model

L283: there's not really a relationship between NMDS dispersion and dietary diversity at the population level; the NMDS is showing dietary differentiation among deer samples not higher diversity in the population diet; of course those two things may be correlated, but deer could still have a highly diverse diet and be completely clumped in the NMDS (e.g. all deer eat the same very diverse diet)

L286-287: I would intuitively agree with the statement that bison are more grazers than deer, but your data do not support this. Graminoids are a tiny fraction of both species' diets and deer eat more graminoids than bison (Supp Fig 1). Given that your samples were collected in summer, when bison should eat the most amount of graminoids according to the literature, this makes your results all the more important, and provides an important counterpoint to the idea that bison are not good browsers.

L296: Fig. 3 does not show dietary diversity. I suggest rephrasing as dietary dispersion

L299: Given than bison in Bialowieza are not grazers (based on your results), does it make sense to attribute this effect to grazing

L304: do bison forage in forest gaps in Bialowieza?

L296-308: alternative explanations for lower intraspecific dietary differentiation in bison. Deer have narrower muzzles enabling them to be more selective when foraging and potentially creating between-individual variation when low-biomass species are eaten; bison have larger digestive systems allowing for more dietary mixing over time and potentially more similarity between individuals; deer are more head-up foragers than bison creating more potential for vertical niche partitioning between deer individuals

L316-317: wouldn't greater overlap be related to behavioral similarities rather than differences?

L320: could mention the role of resource abundance and palatability in determining niche overlap. If there are just a few common plants, it's likely there will be high overlap. If there are a few plants of high quality, both species should eat them

L390-394: I suggest removing these sentences; I do not think you can make this claim without a comprehensive assessment of red deer diet dissimilarity across sites and predictor variables.

6. PLOS authors have the option to publish the peer review history of their article (what does this mean?). If published, this will include your full peer review and any attached files.

Reviewer #1: No

Reviewer #2: No

---

## [Author Response · Author response to Decision Letter 0]

13 Dec 2023

Reviewer #1: The manuscript aims at assessing diet composition and niche overlap within and between red deer and European bison, and correlating such information with habitat quality and predator density, to support management actions. The approach is sound and mostly well explained, I just miss a comparison of the results with information on plant availability in the area and the inclusion of the species movement abilities while discussing the results.

General comments

• I honestly feel that using this approach to estimate predator abundance is an over ambitious goal, since other disturbances and the overall landscape of fear could influence this result. Talking of perception of predator density from the prey species is acceptable, but it should be discussed mentioning the possibility that other factors could also contribute to such perception.

We have now tempered our statements on this matter and have included in the discussion the presence of other factors that are very likely to also have an effect here. Still we wanted to inspire research in this direction so we kept the suggestion in the discussion.

• I missed a parallel with the observed diet composition and plants available in the areas considered. eDNA cannot be used to infer proportions in the diet, however, detecting abundant or scarce species in the diet would provide valuable information in this context.

We have added a whole new paragraph to reflect this point.

• The species movement should be considered when discussing the results, as scats are not deposited where consumption occurred.

We have added a whole new paragraph to reflect this point (same paragraph as previous point).

Specific comments

L36, L109, Fig.1: red deer instead of Red deer.

Adressed

L65-66: this aspect could be better highlighted, since diet composition influences the overall body condition of the species, but also its impact on the vegetation.

Adressed. We have now included body condition as part of the influences from 

L82: probably here the authors meant “study areas”.

adressed

L117: Probably better to replace “It” with “The area” or something alike.

adressed

L121: without forestry nor ungulate management.

adressed

L122: Considering the previous sentence mentions the whole Polish part of the forest, the BNP and the state forest, I’d suggest to replace “It” with a more explicit subject here.

adressed

L149: It seems like “both” refers to predation risk + habitat quality here, but I think the authors likely meant both species, please clarify.

Adressed

L154-156: Where steel beads used in this process? Usually, nitrogen is added when working with plant material, so I was wondering if the authors used the steel beads as recommended instead.

We use silica beads and stored them at room temperature. The amount of silica beads was much greater than the amount of scat to ensure drying. We were unaware of the use of steel beads and nitrogen to store scat samples. Thanks, we will investigate this.

L162-168: the lab procedure is sound and sufficiently explained, I would just suggest to add a reference to the supplementary information and specify the information on the Qubit Kit.

Adressed

L171-172: Please specify the filtering criteria and add a reference to the supplementary information.

Adressed

L174: the filtering in this step is clearly explained and sound.

Thanks

L176-177: this sentence might be a bit challenging for readers with no previous experience with the method, I’d suggest making it a bit more beginner friendly.

L205: Please break down these values per species as well.

Adressed.

Fig.1: is not colour blind friendly. Additionally, please use a larger font for the x-axis.

We chose the colors for the herbiscapes to match the exact same ones as in the eLife paper from Bubnicki et al. 2019. As for the larger x-axis font, we consider it is sufficient, and as it is not truly important for the figure, we don’t want to focus the attention there.

L212-216: please briefly mention the differences among areas as well.

Adressed. We have extended the section on diet comparison using Figure 2B as reference. Thanks for the hint.

Supplementary Fig. 3: Please add test values.

We have removed the figure and the two models as we now consider they don’t contribute much to the manuscript and its better without them.

L237-238: very good point!

Thanks �

L279: “in summer” is not really necessary here.

adressed

L283-284: This is consistent with NMDS results.

L285-288: as mentioned elsewhere, this statement is not fully supported by your results, so it should be better addressed here. Or you could move lines 301-304 here.

Adressed. We have reworked the sentence accordingly.

L304: could also be playing a role.

adressed

Reviewer #2: The authors conduct an interesting study of bison and red deer diets in the Bialowieza ecosystem. They find interesting associations between diet and ecosystem variables and they use appropriate metabarcoding methods. I believe the authors should consider their terminology in some places and consider the value of using both V and overlap. I also believe this is a good study and can be appropriate for publication in PLoS ONE after revisions.

Comments

Abstract: I do not believe that intraspecific dietary dissimilarity is an adequate indicator of predation risk; I think this section of the Abstract should be dramatically tempered. Without a comprehensive survey of possible contributing factors to red deer dietary dispersion (and wolf risk being the only strong predictor of it), it would be disingenuous to recommend that ecosystem managers solely need to measure red deer diets to quantify predation risk.

We have removed the sentence from the abstract. We agree that it can be misleading, but we did not intended the readers to assume that red deer diet is by itself enough to infer predation risk, rather we intended to open the path for this idea to be considered and encourage research in this direction, because we think it has great potential.

L73: there is a parenthesis missing

adressed

L89-98: please add a description of what the herbiscapes are in Bialowieza so the reader has an understanding of Bubnicki et al's results without needing to look up their paper; adding this will help center the reader on your study system and understand what factors go into herbiscapes. Please also add a citation to the Bialowieza-specific study

We agree that it needed a bit more development. It has been added accordingly both in the introduction and in the methods part.

Throughout: I am also a bit uncomfortable with using niche overlap to describe intraspecific diet similarity. I realize that the author's use Pianka's overlap to quantify this so it technically is niche overlap, but I wonder if it might be easier to term the intraspecific values something like "dietary similarity" and reserve the term "niche" for the interspecific comparisons.

We agree that the concept of intra and interspecific overlap might come a bit challenging for some readers. We however would like to have the focus on the overlap and treat both together rather than calling them dietary similarity and niche for intra and interspecific niche overlap.

Methods: The section "Herbiscapes" needs to come before the section "Sample collection" because on L135 the authors state "per area per species" yet the reader has not been told that there are five areas yet

Adressed.

DNA metabarcoding methods: What was the told number of reads per library? It seems quite strict to remove sequences with <100 reads when many studies either remove only singletons or those with <10 reads

We decided to retain only sequences which had occurred 100 times across the whole samples. Thus, this means that we did keep sequences which occurred less than 100 times within samples but not if their cumulative value across samples was less than 100 sequences. As of why 100, we considered that we had enough sequencing power to be this conservative.

Methods: why use V and overlap for within species comparisons? Don't they show the same thing?

They do not show the same thing. The V estimates the diet similarity between an individual and the average diet of its species. In contrast, intraspecific niche overlap reflects the diet differences between two unique individuals, thus only 2 individuals are taken per calculation whereas for V all are taken in account to calculate an average and then compared to each individual.

Fig.2: please add some color to the NMDS; it will make it much easier to distinguish the species

We have changed the color code.

L227: "Food quality" should be "Habitat quality", right?

Yes, thanks for the hint.

L227: The model structure is not completely clear to me; are the categorical predictors nested within species? Is this like having the categorical predictor as a fixed effect and per-species random intercepts?

Adressed. We wanted to nest the quality/risk areas by species so to study them for bison and red deer separatedly. The formulation comes straight from the package used so it might be confusing. We have clarified it in the text.

L238-239: there is only two types of species interactions here (potential competition/facilitation between deer and bison; herbivory between the two species and their food plants). I can guess that the authors are referring to bison intraspecific overlap, deer intraspecific overlap, and deer-bison overlap. If I'm right, please rewrite this part to state that clearly, currently it is not clear.

Adressed and clarified. Thanks.

Modelling methods: I'd like some more justification about the modelling approach for niche overlap. Why include "species interaction" as a predictor rather than just having three separate models (one for each species' intraspecific comparisons and one for interspecific overlap)? This will make it easier to interpret model output. 

We wanted to keep the three types of interactions in the same model so the results could be compared coming from the same exact model formulation. Dividing the model in 3 could have created some biases from the data that we didn’t want to risk, so using species interaction as nesting factor did the job in a clean way.

Regarding the formatting of the model, this again comes from the formatting of the glmmTMB, where we had to include the interaction and the factors separatedly, that is why they show “duplicated” in Model4. We included all the variables in Model 4 so as to be able to predict risk and quality for each type of species interaction separately, rather than for each species interaction, as that was one of our interest points for this research. Since we wanted to account for this interaction variable, we had to keep the three species interactions together within the same model. 

Given that Area is just a proxy for predation risk and habitat quality, why not test those directly from the start (with Area as a random intercept)? It doesn't seem like there is much need for the Overlap ~ Area model

We agree model 3 is not as important and could be removed, but we belive that it provides a good complement and support for the more ambitious model 4.

Regarding model 3, we wanted to have it in order to have an unbiassed pre-view of the niche overlap prediction based on the area itself, to provide contrast for model 4.

L283: there's not really a relationship between NMDS dispersion and dietary diversity at the population level; the NMDS is showing dietary differentiation among deer samples not higher diversity in the population diet; of course those two things may be correlated, but deer could still have a highly diverse diet and be completely clumped in the NMDS (e.g. all deer eat the same very diverse diet)

We agree, thanks for the point. We have reworked the sentence so to clarify it and we have specified your point to distinguish between dietary differentiation and diet diversity.

L286-287: I would intuitively agree with the statement that bison are more grazers than deer, but your data do not support this. Graminoids are a tiny fraction of both species' diets and deer eat more graminoids than bison (Supp Fig 1). Given that your samples were collected in summer, when bison should eat the most amount of graminoids according to the literature, this makes your results all the more important, and provides an important counterpoint to the idea that bison are not good browsers.

Thanks for the point. We have included it to enhance the importance of this result.

L296: Fig. 3 does not show dietary diversity. I suggest rephrasing as dietary dispersion

adressed

L299: Given than bison in Bialowieza are not grazers (based on your results), does it make sense to attribute this effect to grazing

Agreed. We have now addressed this point and reworked the paragraph.

L304: do bison forage in forest gaps in Bialowieza?

Yes, it is quite regular.

L296-308: alternative explanations for lower intraspecific dietary differentiation in bison. Deer have narrower muzzles enabling them to be more selective when foraging and potentially creating between-individual variation when low-biomass species are eaten; bison have larger digestive systems allowing for more dietary mixing over time and potentially more similarity between individuals; deer are more head-up foragers than bison creating more potential for vertical niche partitioning between deer individuals

Adressed. We have included this point in the discussion. Thanks for the hint.

L316-317: wouldn't greater overlap be related to behavioral similarities rather than differences?

Yes, thanks for noticing.

L320: could mention the role of resource abundance and palatability in determining niche overlap. If there are just a few common plants, it's likely there will be high overlap. If there are a few plants of high quality, both species should eat them

Adressed. We have now included this point in the discussion.

L390-394: I suggest removing these sentences; I do not think you can make this claim without a comprehensive assessment of red deer diet dissimilarity across sites and predictor variables.

We have tempered the sentences to acknowledge that more variables are likely to play a role too. However, we would like to maintain the idea to inspire future research in this direction.

---

## [Editor Report · Decision Letter 1]

15 Jan 2024

Niche overlap across landscape variability during summer between two large herbivores using dietary eDNA metabarcoding in the Białowieża Primeval forest (Poland)

PONE-D-22-34041R1

Dear Dr. Dear Dr. Mas-Carrió,

We’re pleased to inform you that your manuscript has been judged scientifically suitable for publication and will be formally accepted for publication once it meets all outstanding technical requirements.

Kind regards,

W. David Walter, Ph.D.

Academic Editor

PLOS ONE

Additional Editor Comments (optional):

I appreciate the authors revisions and the detailed explanation for each point raised by myself and both reviewers.
---

## [Editor Report · Acceptance letter]

1 Feb 2024

PONE-D-22-34041R1 

PLOS ONE

Dear Dr. Mas-Carrió, 

I'm pleased to inform you that your manuscript has been deemed suitable for publication in PLOS ONE. Congratulations! Your manuscript is now being handed over to our production team.

Kind regards, 

on behalf of

Dr. W. David Walter 

Academic Editor

PLOS ONE